# The Activity of Substance P (SP) on the Corneal Epithelium

Jonathan Kopel [1,*] , Caezaan Keshvani [1], Kelly Mitchell [2] and Ted Reid [2]

1   School of Medicine, Texas Tech University Health Sciences Center, Lubbock, TX 79430, USA
2   Department of Ophthalmology and Visual Sciences, Texas Tech University Health Sciences Center,
    Lubbock, TX 79430, USA
*   Correspondence: jonathan.kopel@ttuhsc.edu

**Abstract:** In 1931, Von Euler and Gaddum isolated substance P (SP), an undecapeptide from the tachykinin family, from equine brain and intestine tissue extracts. Numerous types of cells, including neurons, astrocytes, microglia, epithelial, and endothelial cells, as well as immune cells including T-cells, dendritic cells, and eosinophils, are responsible for its production. The corneal epithelium, immune cells, keratocytes, and neurons all express the two isoforms of NK1R, which has the highest affinity for SP. The most recent research supports SP's contribution to corneal healing by encouraging epithelial cell migration and proliferation. Additionally, when applied to the eyes, SP has proinflammatory effects that result in miosis, intraocular inflammation, and conjunctival hyperemia. In this review article, we examine the role of substance P within the eye. We focus on the role of SP with regards to maintenance and healing of the corneal epithelium.

**Keywords:** substance P; cornea; corneal epithelium; ophthalmology

## 1. Introduction

In 1931, Von Euler and Gaddum isolated substance P (SP), an undecapeptide from the tachykinin family, from equine brain and intestine tissue extracts [1]. Numerous types of cells, including neurons, astrocytes, microglia, epithelial and endothelial cells, as well as immune cells including T-cells, dendritic cells (DCs), and eosinophils, are responsible for its production. Neurokinin 1 receptor (NK1R), Neurokinin 2 receptor (NK2R), and Neurokinin 3 receptor (NK3R) are members of the class I (rhodopsin-like) family of G-protein-coupled neurokinin receptors that SP uses to exert its biological effects [1]. The corneal epithelium, immune cells, keratocytes, and neurons all express the two isoforms of NK1R. The trigeminal ganglion ophthalmic branch fibers are primarily responsible for producing SP on the ocular surface. Healthy people's tears have been found to contain SP and its metabolites. SP levels in tears are much lower in people with diabetic keratopathy and corneal hypoesthesia, which disturbs the homeostasis of the ocular surface. In cases of neurotrophic keratopathy, topical use of SP-derived peptide accelerates healing [1]. The most recent research supports SP's contribution to corneal healing by encouraging epithelial cell migration and proliferation. Additionally, when applied to the eyes, SP has proinflammatory effects that result in miosis, intraocular inflammation, and conjunctival hyperemia [1]. In this review article, we examine the role of substance P within the eye. We focus on the role of SP with regards to maintenance and healing of the corneal epithelium.

## 2. Discovery of Substance P

Neuronal and non-neuronal cells generate SP, which is an 11-amino-acid-long neuropeptide that regulates tissue homeostasis, wound healing, and ocular inflammation. Substance P binds to the Neurokinin 1 receptor (NK1R), NK2R, and NK3R, which are G-protein-coupled receptors that SP binds to [2–4]. Among the receptors, NK1R has the most affinity for SP of the three. There is an abundance of data showing that SP controls immune cell activity and microbial infection immunity. Substance P was first detected

in extracts from equine brain and intestine, and was found to induce transient hypotension and to facilitate muscle contraction in the intestine when injected intravenously into anesthetized rabbits. The sequence of SP remained elusive until Leeman and coworkers discovered a peptide that stimulated salivary secretion when injected into anesthetized rats while attempting to isolate corticotropin-releasing factor [5]. This peptide was referred to as sialogen until its physical and chemical characteristics were compared to those of SP. It was later found to be identical to SP. Further purification led to the sequence of SP, reported in 1971 by Chang, Leeman, and Niall: Arg-Pro-Lys-Pro-Gln-Gln-Phe-Phe-Gly-Leu-Met-NH2. Substance P belongs to a family of peptides known as tachykinins. There have been five mammalian tachykinins identified thus far: Substance P, neurokinin A (NKA), neurokinin B (NKB), neuropeptide K (NPK), and neuropeptide Y (NPY). Tachykinins have a common carboxy-terminal sequence Phe-X-Gly-Leu-Met-NH2, where X is an aliphatic or aromatic amino acid [6,7].

## 3. Molecular Biology of Substance P

The nucleic acid sequences for SP, NKA, NPK, and NPY are encoded on a single gene, while the gene encoding NKB lies in a separate location [8–10]. The gene containing the SP sequence is approximately eight kilobases in length and contains seven exons [11]. The transcribed SP encoding gene undergoes alternative splicing to produce three distinct preprotachykinin (PPT) mRNA's that are designated as a-, P- and y-PPT mRNA [11]. A fourth PPT mRNA, 6, has been identified in rats [12]. Even though all PPT mRNA's contain the SP sequence, the mRNAs are subject to differential exon usage. Interestingly, differences in the splicing patterns seem to be tissue-specific [13,14]. In the adult rat, y-PPT mRNA is the most abundant of the three PPT mRNAs. However, in bovines, different PPT mRNAs are more prevalent in different tissues. The exact mechanism for determining differential mRNA splicing is unknown, but the formation of secondary structures in the RNA may be important [15]. These PPT mRNAs, once translated, undergo post-translational processing to form the desired tachykinins.

## 4. Vasoactive Effects of Substance P

The systemic hypotensive effect of SP was one of its defining characteristics and has been shown to result from peripheral vasodilation. When endothelial cells are subjected to physiological changes, such as hypoxia or shear stress, SP is released from these cells [16–19]. Nitric oxide (NO) has been shown to mediate the SP-induced relaxation of porcine coronary arterioles [20]. Jin et al. demonstrated that SP induces vasoactive intestinal peptide (VIP) release and NO production [20,21]. Other results obtained by Sharma and Davis suggest that in porcine coronary artery endothelial cells, SP induces a rise in intracellular $Ca^{2+}$. The resulting increase in calcium activates an intermediate-conductance $Ca^{2+}$-activated K+ channel, which produces hyperpolarization. This ultimately produces a sustained $Ca^{2+}$ entry, which is necessary for endothelium-derived nitric oxide production. SP has also been shown to have profound effects on vasomotion, where a regular, fluctuating flow pattern in skeletal muscles was seen after intravenous injections [22].

In addition, SP causes aggregation of leukocytes and platelets [23]. The effect of SP on vasodilation, vasomotion, leukocytes, and platelets indicates a complex role for this peptide in response to tissue trauma. SP causes a process called neurogenic inflammation, the vasodilation and plasma extravasation that occurs following the release of SP from capsaicin-sensitive nerves [24]. This process was shown to be mediated by the NKl receptor by direct measurement of protein release from blood vessels [24]. Substance P, released by proinflammatory stimuli, causes plasma protein extravasation by increasing endothelial permeability, indicating a potential role for SP in the inflammatory process. The SP antagonist CP-96,345 was shown to block Evans blue dye leakage, a measure of plasma protein extravasation, induced by SP and releasers of endogenous SP [25]. This result indicates that SP controls endothelial permeability by binding at the NKl receptor. Recently SP was also found to cause altered vascular permeability in diabetic rats [26].

## 5. Secretory Effects of Substance P

Substance P may help regulate the release of anterior pituitary hormones. Substance P applied intracerebroventricularly in rats inhibits the release of growth hormone, but increases levels of plasma prolactin [27–29]. Intrathecally applied SP in rats stimulated the release of epinephrine and norepinephrine [30,31]. It has also been suggested that SP interacts with the release of opioid peptides into the circulatory system [32]. In addition, SP has been shown to increase parotid gland secretion through an inositol trisphosphate (IP3)-mediated mechanism, where its effects were measured by the release of labeled protein [33]. Measuring amylase release from dispersed acinar cells has also shown that SP stimulates pancreatic secretion [34]. In the gastrointestinal tract, SP induces secretion of pepsinogen in dogs and inhibition of gastric acid section [35,36]. Contradictory information on insulin and glucagon secretion stems from the fact that SP stimulates the release of these two substances in dogs, but has an inhibitory effect on these substances in rats [37,38]. Substance P also induces histamine release by human mast cells and causes eosinophil cationic protein release by human eosinophils [39,40].

## 6. Mitogenic Effects of Substance P

A new and intriguing development is the discovery that regulatory peptides can also act as mitogens for cells in culture. A direct growth-promoting effect of SP and substance K (NKA) has been reported in smooth muscle cells and human skin fibroblasts [41–44]. Substance P also enhances the proliferation of human blood T-lymphocytes by specific receptors for this peptide [41]. Substance P was also reported to stimulate release of PGE2 and proliferation in rheumatoid synoviocytes, and to stimulate neovascularization [44,45]. Recently, Reid et al. showed that SP at picomolar levels is mitogenic for ocular epithelial cells. These findings are in accord with other evidence which indirectly suggests that tachykinins released by sensory nerves in the skin, joints, and other peripheral tissues may function as mediators of local inflammatory and wound healing responses [46]. It is interesting to note that SP stimulates mitogenesis of embryonic rat aorta cells, but fails to induce significant contraction of these cells. In contrast, SP induced contraction of cultured adult rat vascular smooth muscle cells, but failed to stimulate mitogenesis [47]. Thus, the differentiation state of the cell modulates the mode of action of SP. Substance P may interact with the immune system by partaking in the regulation of lymphocytes. Concentrations in the nanomolar range are capable of stimulating human and mouse lymphocyte proliferation [41]. Substance P has been proposed to be involved in the regulation of glial cell response to injury in the central nervous system [48].

## 7. Muscle Contraction with Substance P Stimulation

Substance P can induce contractions in smooth muscle cells from many different tissues. Muscle strips from guinea pig renal pelvis, rat and guinea pig bladder, rabbit pulmonary artery, rat portal vein, and most parts of the gastrointestinal tract are just some of the tissues that contract in response to SP doses [49–57]. Capsaicin-sensitive primary afferents induce contraction through a direct action on smooth muscle by SP and an indirect action through cholinergic enteric neurons [55,58,59].

## 8. Substance P as a Neurotransmitter

For a compound to be considered a neurotransmitter, three criteria must be satisfied. A neurotransmitter must: (1) be in the proper pre-synaptic structures; (2) be released when appropriately stimulated; and (3) induce the postsynaptic effect or effects in the amounts released. In 1953, Lembeck proposed that SP might be a neurotransmitter. At that time, SP was known to be a vasodilator and able to contract smooth muscle. In addition, several groups had shown that this physiological activity was demonstrated to be much higher in dorsal root tissues rather than in ventral root tissues [60–63]. Substance P was later proven to have an excitatory effect on spinal motor neurons by an extract from bovine dorsal root tissue. When this extract was compared with synthetic SP, the

chemical and pharmacological tests proved to be the same material [64,65]. Several lines of research support the hypothesis of SP as a neurotransmitter. Radioimmunoassay and bioassay concluded that SP levels are approximately 20 times higher in the dorsal horn than in the ventral root [65]. Electron microscopy has determined that SP is associated with large granular vesicles located in primary afferent terminals [66,67]. In addition, immunohistochemistry of rat lumbar dorsal root ganglia revealed that primary afferent C-fibers contain the highest amount of SP immunoreactivity [68]. Stimulation of C-afferent fibers induced an increased release of SP from rat spinal cord [69–71]. Application of SP also produces a dose-dependent depolarization of motor neurons in the spinal cord of newborn rats [64,72,73].

### 9. Ocular Distribution of Substance P

SP is ubiquitous in all ocular tissues, but levels of SP immunoreactivity vary among different species. Elbadri et al. compared SP immunoreactivity levels in cornea, iris/ciliary body, retinal tissues, and choroid/sclera tissues in the cow, sheep, rabbit, and rat [74]. Substance P levels were found to be highest in the retina of cow and rat, whereas in rabbit and sheep, the iris/ciliary body contained the highest degree of SP immunoreactivity. In cat and rabbit retina, most SP-containing cells are located in the ganglion cell layer [75]. Brecha et al. used retrograde labeling from the superior colliculus and optic nerve section to determine that the SP immunoreactive cells in the rabbit retina are ganglion cells [75,76]. Vaney et al., however, determined that the SP immunoreactivity in cats stems from displaced amacrine cells. Immunocytochemistry performed in rats, frogs, lizards, and chicks demonstrated that SP immunoreactivity is prevalent in the amacrine cells of the inner nuclear layer, neurons of the ganglion cell layer, and two distinct layers of processes in the inner plexiform layer [77,78]. These discrepancies in the localization of SP immunoreactivity in the retina may be due to species differences, but the reason for this variability, if it exists, is not known.

In 1984, Osborne presented evidence to implicate SP as a potential neurotransmitter in the bovine retina. He showed that SP binds to retinal membrane preparations with a Kd value of 0.32 nM, suggesting the presence of substance P receptors, and while the retina would not take up exogenous SP, it would release endogenous SP when external $K^+$ concentrations are increased. In more recent studies, Zalutsky and Miller showed that SP at concentrations less than 1 nM excited 78% of rabbit ganglion cells and depolarized some amacrine cells [79]. However, tachykinin antagonists and SP desensitization did not alter characteristics of receptive field properties. Studies conducted by Otori et al. determined that SP levels in the suprachiasmatic nucleus (SCN) are unaffected by changes in lighting conditions or ocular enucleation [80]. Otori et al. also concluded that SP levels in the SCN come from an area other than the retina [80].

Immunoreactivity studies have also determined that the iris, cornea, and ciliary body of several species contain SP [81–86]. Substance P immunoreactive (SPI) fibers enter the cornea from two levels: one from the middle layer of the sclera and the other from the episcleral. From the sclera, a thick SPI fiber trunk, extending to the central part, subdivides into smaller SPI fiber bundles and approaches the epithelium. The SP immunoreactive fiber bundles from the episclera are smaller than those from the sclera. However, both fiber bundles form a dense network in the uppermost part of the stroma [87]. After denervation of the trigeminal nerve, a complete disappearance of all SPI axons in the iris occurs [88]. Blood vessels in the anterior uvea are often surrounded by SP fibers [86]. The SP level in the cornea of the adult mouse is reduced 40% by a similar procedure, and neonatal capsaicin treatment results in an 80% reduction of the SP level in the cornea [89]. Surgical denervation of the trigeminal nerve decreases the level of SP in these areas [82,85,89,90]. This information serves as the basis for believing that the SP levels in the anterior segment of the eye arise from the trigeminal nerve. Murphy et al. reported high concentrations (3 nM) of SP in tears of dogs [91]. The degradation, rather than the release, of SP was recently examined by Igic et al. in rabbit and dog aqueous humor [92]. He concluded that

SP is inactivated by a serine protease in the aqueous humor. A possible serine protease for this function is deamidase, which deamidates SP and other tachykinins [93]. The extensive research on the presence of SP in the eye and the multiple physiological effects of this neurotransmitter make ocular tissue a suitable model for the analysis of SP's interaction with its receptor and pathways involved in SP receptor activation.

## 10. Ocular Smooth Muscle Contraction

The large distribution of SP immunoreactivity in the rabbit and sheep iris/ciliary body gives support to the hypothesis that the physiological role for SP in the eye is to control pupillary diameter [69,94–96]. The iris sphincter muscle is subject to dose-dependent contraction by SP in several species including rabbit, bovine, and pig [97]. Application of tachykinin antagonists reduces the miotic response in these systems [98–100]. Wang and Hakanson, and more recently Kunitomo et al., have shown that the electrically evoked tachykinin-mediated contractile response of the isolated rabbit iris sphincter muscle involves only the SP (NKl) receptor [101]. However, Unger and Tighe reported species differences in the effects of SP on the contractile response of the iris sphincter, and Tachado et al. discovered that while the rabbit, bovine, and pig iris sphincters respond to administered SP by contracting, dog, cat, and human iris sphincter muscles do not [97,102]. However, Anderson et al. contradict this finding by presenting data that demonstrate that human iris sphincter muscle is subject to SP-induced muscle contraction in the eye cup model with an $EC_{50}$ value of 93 nM [103]. Tachado further exemplified these species differences by measuring cAMP formation and IP3 accumulation in these species [97]. The bovine, rabbit, and pig, which were subject to SP-induced muscle contraction, demonstrated a significant increase in IP3 accumulation, but no increase in cAMP formation in the presence of 1 μM SP. Consequently, the dog, cat, and human, which were insensitive to contraction by SP, showed no increase in IP3 accumulation despite a significant increase in cAMP formation. This inter-species variability may suggest that the functional role of SP may be species-dependent.

Other iris smooth muscle studies with non-peptide SP antagonists found that pretreatment with the NKl antagonist CP96,345 produced a right shift in the dose-response curve, with an $IC_{50}$ of approximately 4 mM [104]. While several different laboratories have used different antagonists to implicate the NKl receptor in the SP-stimulated contraction of rabbit iris smooth muscle (Wang and Hakanson used CP96,345; Kunitomo et al. used spantide and L668169; Hall et al. used GR82334), the pA2 (or $IC_{50}$) values were quite high compared with those found for other tissues [101,104,105]. Since the $IC_{50}$ value reported for CP-96,345 binding to rabbit brain is 0.54 nM, compared to the value for the rabbit iris sphincter of 4 nM, these data also appear consistent with the presence of a different NKl receptor subtypes for smooth muscle contraction [106].

Varying data have been reported on the effect of calcitonin gene-related peptide (CGRP) in conjunction with SP upon the induction of miosis [107]. In 1991, Anders et al. summarized the synergistic effects of several neuropeptides with SP. Anders et al. concluded that miosis is due to SP, but not CGRP. Contrary to this information, Andersson and Almegard showed that CGRP fragment 8–37 and fragment 32–37 were functional at inducing a dose-dependent iris sphincter muscle contraction in rabbits [108]. Immunohistochemical analysis of mouse trigeminal ganglia, which express SP, indicated that they also express CGRP. Not all neurons that expressed CGRP contained SP, but the SP-containing neurons also contained detectable CGRP. It is unknown whether CGRP assists in facilitating the functional role of SP in the eye or if CGRP elicits a response of its own that is not mediated by SP. However, the fact that CGRP is found in all of the nerves that contain SP and that CGRP is quite synergistic with SP for cell growth stimulation would seem to suggest that, although CGRP may function alone, it probably also functions in association with SP [109].

## 11. Substance P and Corneal Pain

The cornea is the most sensitive and innervated tissue on the ocular surface. Myelinated and unmyelinated sensory fibers sandwiched between the various layers of the corneal epithelium provide the only innervation to the cornea. Therefore, preserving the integrity of the cornea from potential damage remains important for maintaining the integrity of the eye structure and function. Specifically, corneal mechanical, chemical, or thermal stimulation produces aversive or nociceptive sensations, except for the purely cold sensations elicited by low-temperature stimuli [110–115]. Moreover, direct corneal nerve terminal stimulation triggers defensive reflexes such tear production, eye blinking, and endocrine and cardiovascular responses [110–115]. The trigeminal nerve (ophthalmic branch, V1), whose ganglion includes the somas of the primary sensory neurons, conducts primary afferent fibers that innervate the cornea. These fibers are of the myelinated A and unmyelinated C types. The trigeminal spinal nucleus (Sp5) within the brainstem acts as the center, whereby the central branches of corneal afferents touch the second order sensory neurons, which are represented by both projection and local circuit neurons. Many ophthalmic and systemic illnesses have a deleterious effect on the structure and function of the corneal nerve. A long-lasting noxious stimulus, injury to the ocular surface, ocular neurosensory abnormalities, or damage to the ocular surface can all cause persistent ocular pain (neuropathic pain). Pain sensitization, which can show as spontaneous pain, hyperalgesia, and allodynia, can be brought on by persistent and aberrant activation of corneal nociceptors [116,117]. Damaged corneal tissue and immune cells release several molecules and inflammatory mediators that interact with the membrane receptors/channels of the nociceptor ending membrane, including ATP, hydrogen ions, SP, neurokinin A, tumor necrosis factor alpha, prostaglandin E2, and interleukins.

Trigeminal fibers release neuropeptides, which have well-known roles in neuroinflammatory processes and the control of nociceptive signal processing. Many effects of SP include neurogenic inflammation and nociception (pain perception) [118–126]. According to current research, diabetic mice have significantly fewer SP nerves than wild-type mice, which accounts for around 59% of the total epithelial innervation in the mouse cornea [118–126]. An ion channel known as transient receptor potential melastatin (TRPM8) is found in sensory neurons and functions as a key cold and osmolarity sensor [118–126]. Several studies have shown that TRPM8 regulates the wetting of the ocular surface and that altered expression of the TRPM8 channel contributes to cold allodynia and neuropathic pain [118–126]. It has recently been shown that abnormal SP expression, low concentration of TRPM8 at nerve terminals, and hypersensitive nerve response occur long after the injury, and changes in gene expression in the trigeminal nerves may contribute to the pathogenesis of corneal surgery-induced dry-eye-like pain [127].

## 12. Substance P and Corneal Epithelium Healing

One of the first experiments to demonstrate the link between SP and corneal epithelium healing was done by the French physiologist, François Magendie, while examining patients with neurotrophic keratopathy or "neuro-paralytic keratitis" [128]. Neurotrophic keratitis is a rare degenerative corneal disease characterized by lack of or decreased corneal sensation, corneal epithelial breakdown, and impaired healing, resulting in increased susceptibility of the corneal surface to injury and compromised healing. Without treatment, this can lead to stromal melting, corneal ulceration, and corneal perforation. While documenting and treating patients with neurotrophic keratopathy, Magendie hypothesized that the presence of trophic nerve fibers in the trigeminal nerve may be important for regulating tissue metabolism and healing. Subsequent studies later identified the substance involved in the pathogenesis of neurotrophic keratopathy and corneal healing was SP [128].

Over the last few decades, there have been several pre-clinical and clinical studies to show the importance of SP with corneal epithelium healing. Bee et al. showed that collagen type IV intrastromal fibers are orthogonal to the epithelial basement membrane in the cornea [129]. The neurons that innervate the epithelium display SP immunoreactivity

abundantly. On the twelfth day of development, SP immunoreactive nerves are first discovered, coinciding with the beginning of epithelial innervation rather than the extension of nerves through the stroma [129]. Such nerve fibers exhibited substantial connection with both basal and superficial epithelial cells, and increase in number as the body develops. Therefore, SP primary afferents are abundantly supplied to the avian cornea [129]. Furthermore, the expression of SP immunoreactivity correlates directly with the initiation of innervation of the corneal epithelium. Another study by Miller et al. found a dense network of SP immunoreactive axons in the cornea's substantia propria, subepithelial layer, and corneal epithelium [82].

After the superior cervical ganglion was removed, most of the SP immunoreactive axons in the iris developed. Therefore, Miller et al. determined that sensory neurons with SP positivity supply the iris and cornea [82]. A subsequent study by Tervo et al. examining human corneas and irises obtained from corneal surgery or sector iridectomies showed that varicose fibers of the corneal epithelium and nerve trunks in the corneal stroma both exhibited SP immunofluorescence [85]. Substance P immunoreactive nerve fibers in the iris were primarily found near the pupillary edge in both dilator and sphincter areas [85]. Additional radioisotope studies further confirmed that specific binding of SP was found in the iris sphincter muscle, choroid, and retina [130]. In rats, detectable amounts of SP-binding sites were also expressed in the corneal epithelium and iridial stroma [130]. Subsequently, it was discovered that SP has important functions in corneal re-epithelialization after injury (Figure 1) [91,131,132].

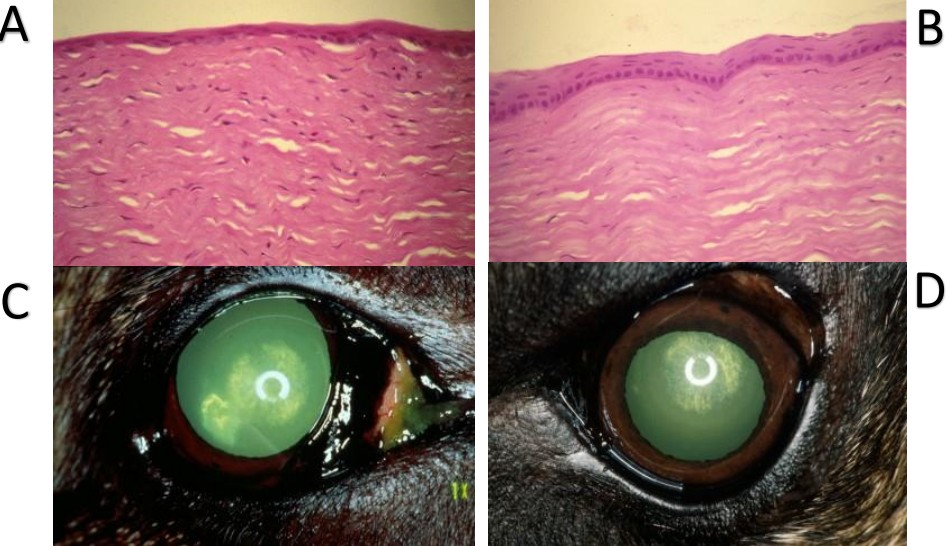

**Figure 1.** Corneal healing using SP in dogs. (**A**) Cornea of capsaicin rabbit after healing—not treated with Substance P (note epithelium is not well attached). (**B**) Cornea of capsaicin rabbit after healing—treated with Substance P (note epithelium is well attached). (**C**) Dog eye with corneal ulcer in the lower left corner (haze in the background is a cataract). (**D**) Dog cornea after one week treatment with substance P, which shows resolution of the corneal ulcer.

Further investigation demonstrates that neither SP nor IGF-1 alone affects corneal epithelial wound closure in vivo, but that they act synergistically to stimulate corneal re-epithelialization by activating the NK1 receptor [133–135]. Specifically, Nishida et al. showed that the addition of either SP or IGF-1 alone did not affect epithelial migration, while the combination of SP and IGF-1 stimulated epithelial migration in a dose-dependent fashion [136]. Their study also showed that the synergistic effects of SP and IGF-1 on corneal epithelial migration were inhibited using an SP antagonist or enkephalinase [136]. Interestingly, the combination of SP and IGF-1 did not affect the incorporation of 'H-thymidine into corneal epithelial cells [136]. Instead, Nishida et al. showed that SP with

IGF-1 caused an increase in attachment of corneal epithelial cells to fibronectin, collagen type IV, and laminin matrices [136]. A similar improvement in corneal epithelial healing was observed when an NK-1 receptor agonist was used, which suggests that the synergistic effect of SP and IGF-1 might be mediated through the NK-1 receptor system [136]. These results suggest that the maintenance of the normal integrity of the corneal epithelium might be regulated by both humoral and neural factors [136].

Another study by Ofuji et al. suggested that other pathways, namely the tyrosine kinase and protein kinase C, may also be involved in corneal healing with SP activation [137]. Once activated, SP induces corneal healing through downstream effects in the integrins, zonula occludens-1, focal adhesion kinase, paxillin systems, E-cadherins, calmodulin-dependent protein kinase II, and p38 mitogen-activated protein kinase, all of which have important roles in epithelial remodeling and healing [138–142]. This follows closely with a previous study by Kingsley et al., which showed that topical application of SP or its NK1 receptor antagonist has no significant effect on the rate of corneal epithelial wound closure in the rabbit [143]. However, these were normal rabbits and nothing had been done to deplete their normal reserves of SP found in their corneal sensory nerves. The effects of SP on corneal epithelial healing was demonstrated in both mouse model and human studies, which showed improvements in neurotrophic keratopathy, postsurgical superficial punctate keratopathy, and recurrent corneal erosion in diabetic patients with native SP and SP analogs, such as phenylalanineglycine-leucine-methionine amide (FGLM), in conjunction with insulin-like growth factor-1 [144–154]. An example of the effects of Substance P corneal healing is shown in Figure 2 [155]. The patient in Figure 2 was born with no corneal sensory nerves and a severe corneal ulcer in the right eye, which was about to perforate. A skin flap was pulled to cover the right eye. In addition, the patient later developed a corneal ulcer on the left eye that was also close to perforating. Patient was treated with SP plus IGF-1 in an eye drop twice daily (one drop every 15 min for two hours in the morning and at night. The drops contained normal saline). Substance P is added over two hours since it must be present in the wound for that time. The patient showed complete healing of the ulcer in two weeks. Other studies found that adjunct therapies, such as tetrapeptide (SSSR; Ser33-Ser-Ser-Arg) derived from the C domain of IGF-1 and capsaicin, improved both corneal epithelial along with SP and IGF-1 [156–158]. Through the activation of Akt and ROS scavenging through the NK-1 receptor, SP protects corneal epithelial cells from the apoptosis that is brought on by hyperosmotic stress [159].

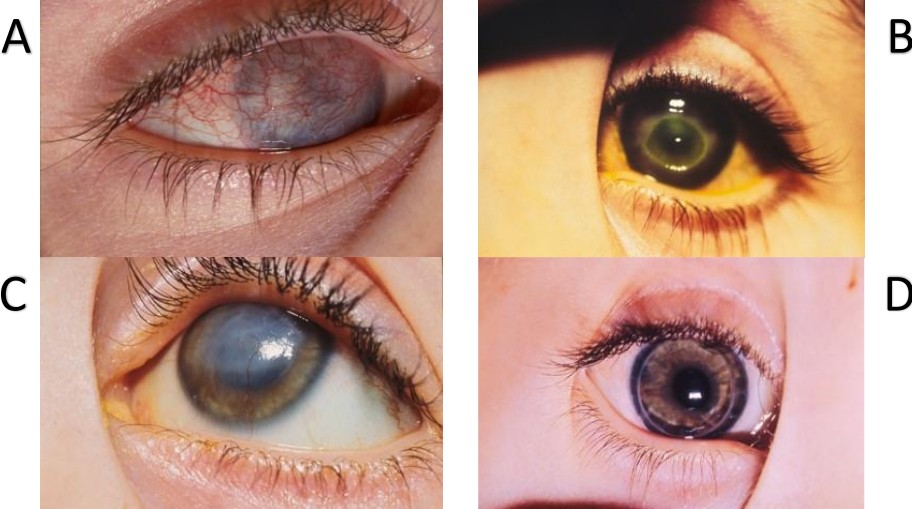

**Figure 2.** The effect of Substance P and IGF-1 on corneal ulcer healing in a human patient. (**A**) Two-year-old patient with skin flap over ulcer in right eye. (**B**) Same patient with large corneal ulcer in left eye. (**C**) Left eye after two weeks treatment with SP with no ulcer remaining. (**D**) Right eye, six months after cornea transplant and treatment with SP.

### 13. Therapeutic Applications of SP in Cornea Epithelium

Currently, SP research has created several NK-1R/substance P antagonists. Research is underway on substance P/NK1R antagonists as antidepressants, anxiolytics, and anti-inflammatory medications [160]. The first studies on NK-1R/substance P antagonists were originally tested as antidepressants, but subsequent research revealed them to have antiemetic effects. Specifically, NK-1R/substance P antagonists were shown to be effective antiemetic drugs for chemotherapy-induced vomiting by inhibiting substance P from binding to NK1 receptors in the region postrema, which regulates emesis. NK-1 receptor antagonists, such as aprepitant and its prodrug fosaprepitant, are utilized as antiemetic medications. Both oral and intravenous (IV) delivery of aprepitant are accessible; however, only the intravenous version of aprepitant is available. These two medications are helpful in preventing nausea and vomiting brought on by chemotherapy. Hepatic enzymes in the body transform fosaprepitant, a prodrug of aprepitant, into aprepitant, the chemical that is physiologically active [161,162]. Netupitant is also used as an antiemetic in conjunction with Palonosetron as a preventative measure before chemotherapy to lessen nausea and vomiting [163,164]. It has been demonstrated that the chile pepper compound capsaicin reduces the concentration of SP at the terminal and peripheral nerve ends of afferent neurons. Specifically, capsaicin reduces the perception of painful stimuli because of substance P's function in pain transmission [165–167]. Nerve growth factor, a chemical required to produce substance P, is interfered with by capsaicin. Arthritis, post-herpetic neuralgia, shingles, fibromyalgia, and peripheral diabetic neuropathy are treated with capsaicin cream to reduce pain [168–170].

As described in the previous sections, SP has been shown to be an important component for neurogenic inflammation, as it can modulate functions of various immune cells like microglia and dendritic cells [152,159]. Substance P has been shown to be important in aspects of chronic pruritus and was shown to activate Mas Related G-protein-coupled receptor MrgprX2, present on mast cells, besides NK1 receptor [171]. Since SP has been shown to activate NK1 receptors, aprepitant, serlepitant, and other NK1 antagonists have been investigated in clinical trials in treating chronic severe pruritus [172]. In the cornea, SP has been shown to cause avascular hemangiogenesis and lymphangiogenesis in mice models. Topical application of NK1 antagonist, Lanepitant, was shown to reduce the angiogenesis within the inflamed cornea, and this was also shown to occur with topical Fosaprepitant [173,174]. In a recent paper, ocular mast cells were also shown to cause increased inflammatory angiogenesis or neovascularization due to release of VEGF-A in mouse models of corneal inflammation [175]. They were able to show inhibition of mast cells activation resulting in decreased angiogenesis. In Figure 3, we summarize the potential role of SP in different regions of the eye where it is expressed with possible functions.

In mouse models of *Pseudomonas* corneal infection, it was shown that high levels of SP caused an increase in inflammatory cytokines, resulting in more adverse reactions in the cornea. The same group also showed that using NK1 antagonist Spantide 1 improved outcomes in the *Pseudomonas*-infected cornea since the levels of cytokines were reduced due to inhibition of SP-mediated NK1 activation [176]. Since SP mediates angiogenesis via NK1 activation in cancer, NK1 antagonists have been studied in animal models of hepatoblastoma and small cell lung cancer as potential therapeutic options to reduce tumor burden using in vivo and in vitro models due to their anti-angiogenic effect as well as potential for increasing apoptosis [177,178]. There have also been cases where topical application of SP was able to show corneal epithelial wound healing in diabetic mice models as well as model of alkali burn in mouse and rabbit eyes [152,179]. In 1997, there was a study to test topical application of SP on rabbit eyes for wound closure which did not show any statistical significance in wound closure; however, these were normal rabbits and nothing had been done to deplete their normal reserves of SP found in their corneal sensory nerves [143]. In the section above, we discussed the beneficial effect of FGLM-amide peptide with IGF1 in corneal neurotrophic keratopathy. Currently there are no therapies of direct topical SP application for corneal healing that are approved by the FDA. Most substance

P antagonists (e.g., Spantide-1 and -2, Lanepitant, L-733,0660, L-732,138, L-733,060, and L-732,138) have been tested in vitro or in vivo [180]. These substances have demonstrated several effects on reducing inflammation, apoptosis, hemangiogenesis, corneal sensitivity, and cell migration. However, further clinical studies are needed to assess the efficacy and safety in human patients [180].

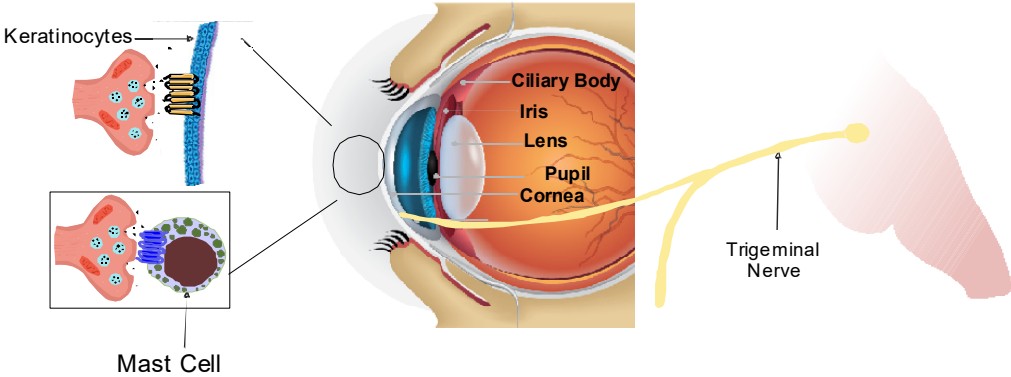

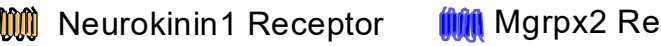 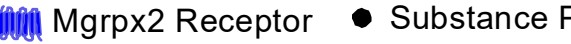

**Figure 3.** Substance P release from trigeminal nerve: Trigeminal nerve (V1) innervates the corneal epithelium, and it releases SP from its axonal terminals. Keratinocytes have been shown to express Neurokinin 1 (Nk1) receptors that promote inflammation and are important for wound regeneration. Mast cells have been shown to express MgrpX2 receptor that stimulates mast cells to release cytokines and promote inflammation.

## 14. Conclusions

Substance P is an important neuropeptide that results in its effects via activation of neurokinin receptors as well as MgprX2 receptors [180]. The current field of study in ophthalmology for SP has been progressing to target SP combined with IGF1 and other molecules to promote wound healing in damaged corneas. On the other hand, efforts are underway to use SP antagonists that prevent binding of SP to its target effectors in controlling inflammation and neovascularization for various pathologies, including keratopathies. As we learn more about the complex nature of SP in various organ systems and the results from clinical trials in different fields like dermatology, they will guide us to start potential trials using SP or its antagonists in helping patients with corneal inflammation and wound healing.

**Author Contributions:** Conception and design: J.K., C.K. and K.M. Manuscript writing: J.K., C.K. and K.M. Final approval of manuscript: T.R. All authors have read and agreed to the published version of the manuscript.

**Funding:** This research received no external funding.

**Institutional Review Board Statement:** Not applicable.

**Informed Consent Statement:** Not applicable.

**Data Availability Statement:** Not applicable.

**Conflicts of Interest:** The authors declare no conflict of interest.

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
