# Peer review of "The Activity of Substance P (SP) on the Corneal Epithelium"

_2813-1053, doi:10.3390/jcto1020006_

Round 1

Reviewer 1 Report

1. The wound healing effect of SP is well described, including Figure 2. Are there any other reports about corneal wound healing effect of SP?

Question 2. SP and corneal pain sensitivity: what is known? Adding a paragraph about this information would be helpful to understand the role of SP better. 

Author Response

Reviewer 1 1. The wound healing effect of SP is well described, including Figure 2. Are there any other reports about corneal wound healing effect of SP? -We appreciate the reviewer’s comments. There are no other papers we could find that report on corneal wound healing effect of SP beyond what we mention in this manuscript. Question 2. SP and corneal pain sensitivity: what is known? Adding a paragraph about this information would be helpful to understand the role of SP better. -We appreciate the reviewer’s comments. We added two paragraphs to the manuscript to expand on the role of SP in corneal pain sensitivity.

Reviewer 2 Report

This is a well-written review article on functions of substance P. 

The reviewer would like to suggest to refer the first publication on the synergistic effects of substance P and IGF-1 in organ culture of the cornea (Nishida, T. et al. J Cell Physiol 169, 159-66, 1996) in Line 296-298.

Author Response

Reviewer 2

This is a well-written review article on functions of substance P. 

-We appreciate the reviewer’s comment on the manuscript.

The reviewer would like to suggest to refer the first publication on the synergistic effects of substance P and IGF-1 in organ culture of the cornea (Nishida, T. et al. J Cell Physiol 169, 159-66, 1996) in Line 296-298.

-We appreciate the reviewer’s comment. We expanded lines 296 – 298 to include details from the Nishida et al. study.